# Electrochemical Oxidation Profile of Anthocyanin Keracyanin on Glassy and Screen-Printed Carbon Electrodes

**Emad F. Newair *** [ID]**, Aboelhasan G. Shehata and Menna Essam**

Unit of Analytical Chemistry (UAC), Chemistry Department, Faculty of Science, Sohag University, Sohag 82524, Egypt; aboelhassan2016005@science.sohag.edu.eg (A.G.S.); menna_2100441@science.sohag.edu.eg (M.E.)
* Correspondence: emad.newair@science.sohag.edu.eg

**Abstract:** A study of keracyanin chloride (KC) electrochemical behavior in an aqueous buffer solution using screen-printed carbon electrodes (SPCEs) and glassy carbon electrodes (GCEs) was performed. Cyclic voltammetry (CV) and square-wave voltammetry (SWV) were used to analyze the electrochemical response of KC under studied conditions. A clear redox wave was observed for KC, primarily due to the oxidation of the catechol 3′,4′-dihydroxyl group of its ring B, with a minor redox wave from oxidation of the hydroxyl groups in ring A. Compared to GCEs, using modified SPCEs resulted in two-fold amplification in the electrochemical oxidation signal of KC. Using SPCEs as a working electrode could provide high sensitivity in the quantification of KC and the ability to gauge KC quantification to significantly lower detection limits.

**Keywords:** keracyanin; antioxidant; voltammetry; electrochemical sensor; screen-printed electrode

## 1. Introduction

Anthocyanins are pigments found in the vacuoles of higher plants that are water-soluble and biologically active. In fruits and vegetables, legumes, and cereals, anthocyanin pigments are thought to be the most abundant water-soluble pigments [1,2]. Recent epidemiological studies suggest that eating fruits and vegetables regularly reduces the risk of aging-associated chronic diseases [3–5]. In addition to their many health benefits [6–8], anthocyanins are antioxidant, antitumor, antiradical, antimutagenic, antiproliferative, antiapoptotic, and nitric oxide-inhibitory [9–15]. Anthocyanins are flavonoids with different structural characteristics, such as the number and nature of hydroxyl groups, the degree of methylation of -OH groups, and sugar attachments [16].

Anthocyanin keracyanin chloride has significant antioxidant properties due to the breakdown of the O–H bonds attached to the aromatic ring (not the glycoside OH). Researchers have demonstrated that anthocyanins with a lower oxidation potential and are more efficient at scavenging radicals [17,18]. Antioxidant capacity and electrochemical behavior have a direct relationship: the lower the oxidation potential, the greater the antioxidant capacity [19]. A voltammetric signal at low anodic potentials indicates the presence of polyphenolic compounds with high antioxidant capabilities, while an oxidation signal at high potentials indicates the presence of polyphenolic compounds with low antioxidant capacities [20]. Studies involving voltammetric measurements of fruits are rare, and only those involving blackberries and raspberries were found [21,22]. The anthocyanin content and radical scavenging capacity of non-Vitis vinifera grapes were determined using spectrophotometry, HPLC with electrochemical detection, and matrix-assisted laser desorption ionization [23]. Electrochemical sensors derived from sewage sludge were used to detect anthocyanin in berry fruits [24]. Anthocyanin contents and the antioxidant capacity of grapes were measured using screen-printed carbon electrodes modified with single-walled carbon nanotubes (SWCNT-SPCE) [25]. As a result, it is necessarily better to understand the electrochemical properties of anthocyanin keracyanin chloride to appreciate its antioxidant

capacity fully. The B ring is reported to be more oxidizable than the A ring (Figure 1) [26]. This study aims to examine the electrochemical oxidation behavior of keracyanin chloride since this behavior has never been studied before. We used different electrode materials, including glassy carbon and screen-printed electrodes, to investigate keracyanin chloride's electrochemical oxidation behavior.

**Figure 1.** Chemical Structure of keracyanin chloride (cyanidin-3-*O*-rutinoside chloride).

## 2. Materials and Methods

### 2.1. Chemicals and Solutions

Merck supplied boric acid 99.95%, glacial acetic acid 100%, and phosphoric acid 85%. Britton–Robinson buffered solution (B-R) was prepared from 0.04 M acetic, boric, and phosphoric acids. Additionally, the buffered solution of B-R was adjusted for pH values by adding 0.2 M sodium hydroxide solution until a pH value of 2.2 was achieved. A water/ethanol (50:50, %*v/v*) mixture was used to prepare a stock solution of keracyanin chloride (KC). All experiments were conducted using significantly bi-distilled water to prepare fresh solutions.

### 2.2. Instruments

The electrochemical experiments were conducted on PGSTAT128N Autolab Electrochemical Workstation powered by NOVA 2.0 software (current resolution is 0.0003%, and the current accuracy is 0.2% of the current range) (Eco-Chemie, Utrecht, The Netherlands). High-purity nitrogen was used for degassing the solution before electrochemical measurements. Afterward, nitrogen was used as a blanket. All experiments were conducted at room temperature. A pH meter HI2221 (Hanna Instruments, Bucharest, Romania) was utilized to adjust the pH of solutions.

### 2.3. Disk Electrode

The study used three electrodes: a glassy carbon electrode (GCE, 3 mm diameter with a surface area of $0.07$ cm$^2$) for working electrodes, an Ag/AgCl electrode for reference electrodes, and a platinum electrode for counter electrodes. Two minutes were spent polishing the glassy carbon electrode with aqueous suspensions of 1.0 microns alumina and 0.05 microns diamond before each run. After rinsing with distilled water, the GCE was sonicated in ethanol and distilled water for five minutes. With a potential range of $-1$ to $+1$ V, cyclic voltammetry was used to clean the GCE electrochemically for 25 cycles at a scan rate of 50 mV s$^{-1}$ in B-R buffer solution (pH 2.2, KC becomes more hydrophobic in an acidic medium due to their nonionizing OH groups, which migrate on the hydrophobic surface of GCE [20]).

### 2.4. Screen-Printed Electrodes

Silver reference and carbon counter electrodes were also used with a screen-printed working electrode (4 mm diameter, DropSens, Spain). Three screen-printed electrodes were utilized, including screen-printed carbon electrode (SPCE), single-walled carbon nanotube-modified SPCE (SWCNTs-SPCE), and multi-walled carbon nanotube-modified

SPCE (MWCNTs-SPCE). The electrode was electrochemically measured after casting 200 µL of a keracyanin chloride solution onto it.

## 3. Results and Discussion

### 3.1. Voltammetric Behavior of KC on GCE

CV analysis of 68 µM keracyanin in 0.04 M B-R buffer (pH 2.2) was performed on GCE at a scan rate of 100 mV s$^{-1}$ in a potential window between 0 and +1.4 V (Figure 2, red voltammogram). One oxidation peak was observed at +0.59 V on the voltammogram, while two oxidation shoulders were observed at +1.1 V and +1.2 V. The anodic peak at +0.59 V indicates the formation of *o*-quinone through the oxidation of catechols in the B ring. In contrast, the smaller shoulders indicate the oxidation of hydroxyl groups in the less electroactive ring A [27,28]. The behavior of KC in the oxidation process was similar to what was observed previously in procyanidin B2 [27]. A previous study reported that the hydroxyl groups of the catechol B ring are more easily oxidized than those of the resorcinol A ring [29,30]. When the potential scan is switched at +0.9 V, the anodic peak at +0.59 V indicates a reversible reaction (inset in Figure 2, blue voltammogram). It is illustrated in a potential window between +0.2 and +0.9 V that KC illustrates a redox wave with an anodic peak potential ($E_p^a$) of +0.58 V and a cathodic peak potential ($E_p^c$) of +0.5 V. It has been shown that the peak separation was 80 mV when measured $\Delta E_p = (E_p^a - E_p^c)$, which is a higher value than expected in a fully reversible system with two electrons (59/n mV). Moreover, it should be noted that the peak current ratio ($i_p^c / i_p^a$) was less than unity. In the inset in Figure 2, the black voltammogram indicates that the background buffer does not exhibit a redox response without keracyanin. As a result of these data, it can be concluded that the redox process of KC at GCE was quasi-reversible. The anodic peak can also be used for electrochemical sensing of KC in the future.

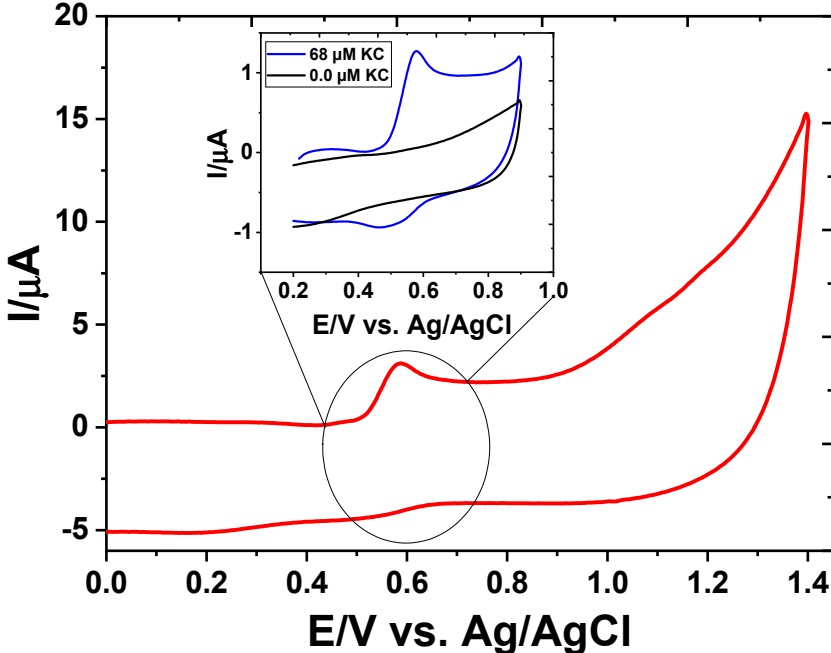

**Figure 2.** Cyclic voltammograms (CVs) of 68 µM keracyanin in 0.04 M B-R buffer (pH 2.2) on GCE at scan rate 100 mV s$^{-1}$. Inset: blue voltammogram of 68 µM keracyanin in a small potential window from +0.2 to +0.9 V, and black voltammogram is the background current in the absence of KC.

SWV was also used on GCE to investigate the electrochemical behavior of KC in 0.04 M B-R buffer (pH 2.2). SWV is more efficient than CV due to its faster analysis speed, lower redox species consumption, and more minor electrode surface poisoning problems [31]. Figure 3A shows the SWV of 68 µM keracyanin in 0.04 M B-R buffer (pH 2.2) on GCE.

Interestingly, a new oxidation shoulder is observed at +0.45 V, and the original anodic peak is found in the same position (+0.59 V). It is possible to explain this because SWV is more susceptible to the enhancement in the oxidation of the catechol moiety of ring B by two oxidation waves.

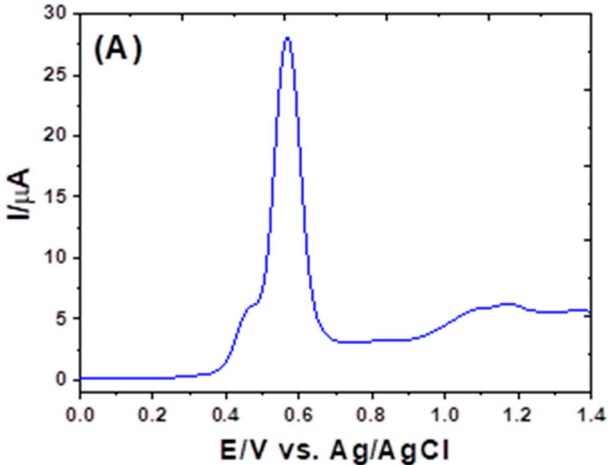 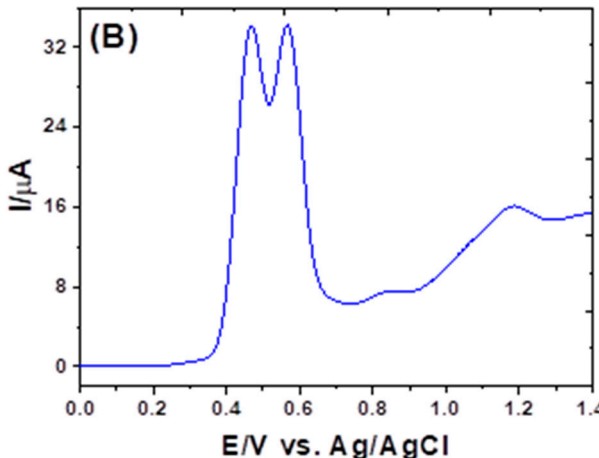

**Figure 3.** Square wave voltammogram (SWV) of 68 μM keracyanin in 0.04 M B-R buffer (pH 2.2) on GCE at frequency 20 Hz (**A**) non-adsorptive stripping SWV and (**B**) adsorptive stripping SWV with deposition potential 0.2 V for 60 s accumulation time.

An electroanalytical method can be made more sensitive by employing several strategies. In general, voltammetry is selected as the technique of choice, an accumulation step is incorporated, and the electroactive species are eventually regenerated through a catalytic chemical reaction. As a preconcentration step, electrochemical or adsorptive methods are typically used in electroanalytical applications. Normally, electroactive species are accumulated on the working electrode through electrochemical processes or adsorptive processes. As soon as the electroactive species have accumulated, the potential is scanned, and the current is sampled. The square-wave voltammetry (SWV) technique is widely used to analyze electrochemical variables quantitatively. Both oxidation peaks of KC were enhanced when adsorptive SWV (Figure 3B) was applied at a deposition potential of 0.2 V for 60 s accumulation time. The adsorptive SWV enhanced the other two shoulders involved in ring A hydroxylation at higher positive potentials. The difference in the measured current of KC between SWV and CV is approximately ten-times greater when SWV is used instead of CV. In this regard, square wave voltammetry is one of the most sensitive techniques for measuring current responses.

The voltammetric oxidation of KC was performed for various pHs of the supporting electrolyte on GCE. pH values were between 1.8 and 9.9, and the concentration of KC was fixed to 20 μM CFT in 0.04 M B-R buffer solution (Figure 4A). The anodic peak of KC oxidation was observed at all pH levels, and its position varied with pH, providing valuable information regarding the electrode process. The peak potential of oxidation ($E_p$) variation versus pH can be observed in Figure 4B, which shifts linearly towards lower positive values until the highest pH value (pH 9.9). This means that the KC oxidation mechanism is clearly impacted by deprotonation. Figure 4B shows a linear relationship for $E_p$ vs. pH: $E_p = 0.68 - 0.06$ pH. Therefore, KC is electrochemically oxidized to produce the corresponding *o*-quinone by two-electron, two-proton oxidation, as shown in Scheme 1 [32].

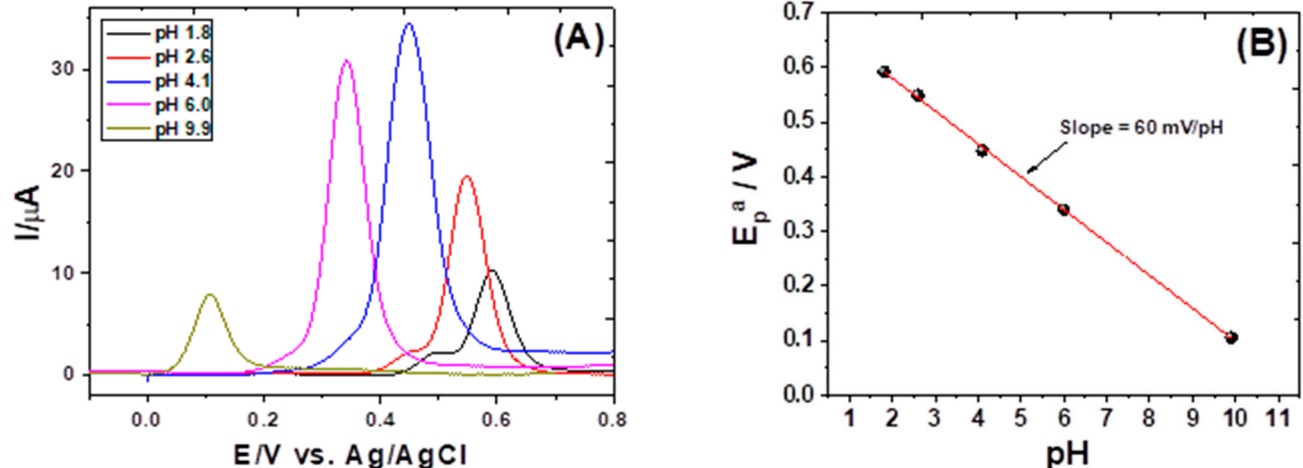

**Figure 4.** (**A**) Square wave voltammogram (SWV) of 20 μM keracyanin in 0.04 M B-R buffer at GCE at frequency 20 Hz, pulse amplitude 25 mV and potential step 1 mV at different pH values, (**B**) $E_p$ vs. pH relationship.

**Keracyanin (KC)**　　　　　　　　　　　　　**KC *o*-quinone**

**Scheme 1.** The oxidation mechanism of keracyanin (KC).

Using the bare GCE, square wave voltammetry (SWV) was used to produce the calibration curve for KC. According to Figure 5, square wave voltammograms were obtained at various KC concentrations. As a result of the redox process associated with KC, a well-defined wave can be observed for all voltammograms near +0.5 V. The inset in Figure 5 illustrates the graph plotting KC concentrations against the peak areas. According to the regression analysis, the linear equation obtained was observed from 10 to 60 m KC with a correlation coefficient of 0.996 as follows:

$$\text{Peak Area (CV/s)} = 3.87 \times 10^{-7} + 0.0726\,[\text{KC}]\,(\mu\text{M}), \text{ r} = 0.996$$

In order to calculate the peak area, the SW peak is integrated from 0.3 to 0.75 V to obtain a unit of an ampere voltage (A.V). Since ampere equals charge (C) per second (s), the final unit of the peak area is always calculated as (CV/s). There is a linear relationship between the concentration of the analyte and the area under the peak. Accordingly, the present analytical method was found to be linear within the specified range of values. Analytical curve parameters were used to calculate the limit of detection (LOD). LOD = 3 $S_b$/s, where $S_b$ is the standard deviation of the y-intercept and s is the slope. Based on the given conditions, the calculated LOD is 5.2 μM.

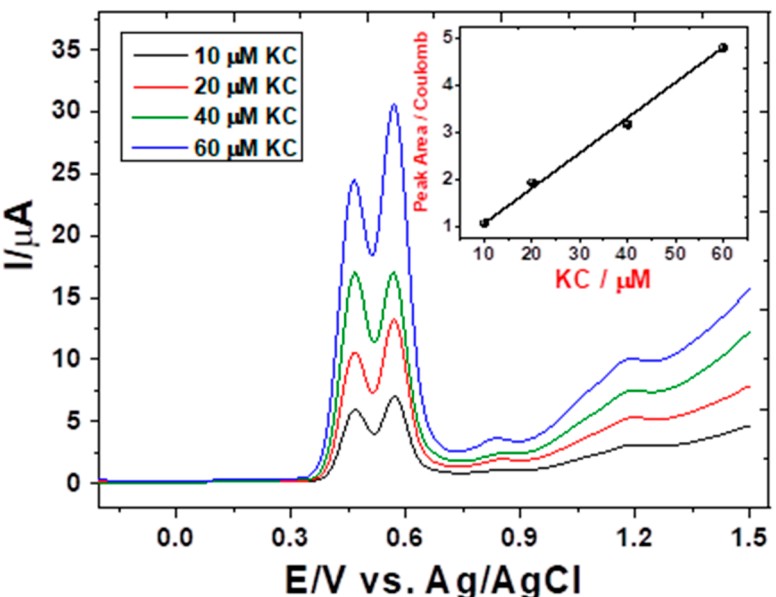

**Figure 5.** Adsorptive square wave voltammograms (AdSWVs) of keracyanin in B-R buffer (pH 2.2) at GCE at deposition potential 0.2 V, accumulation time 60 s, frequency 20 Hz at different KC concentrations.

### 3.2. Voltammetric Behavior of KC on SPCEs

The electrochemical response of keracyanin chloride was also determined using unmodified screen-printed carbon electrodes (SPCEs), single-walled carbon nanotube-modified screen-printed carbon electrodes (SWCNTs-SPCEs), and multi-walled carbon nanotube-modified screen-printed carbon electrodes (MWCNTs-SPCEs). A square wave voltammogram of 68 µM keracyanin in 0.04 M B-R buffer (pH 2.2) at screen-printed carbon electrode (SPCE) (Figure 6) at a frequency of 20 Hz in a potential range of −0.5 V until +1.6 V vs. Ag illustrates similar oxidation behavior obtained on GCE. The square wave voltammograms using SPCEs during electrochemical experiments present one main anodic peak with a corresponding potential of around +0.35 V, with one shoulder at around 0.24 V vs. Ag, corresponding to the oxidation of the 3,4-hydroxyl groups of B ring to form the corresponding *o*-quinone [33]. Two more shoulders at a potential of around +0.96 V and 1.4 V vs. Ag correspond to the oxidation of the hydroxyl groups of the A ring.

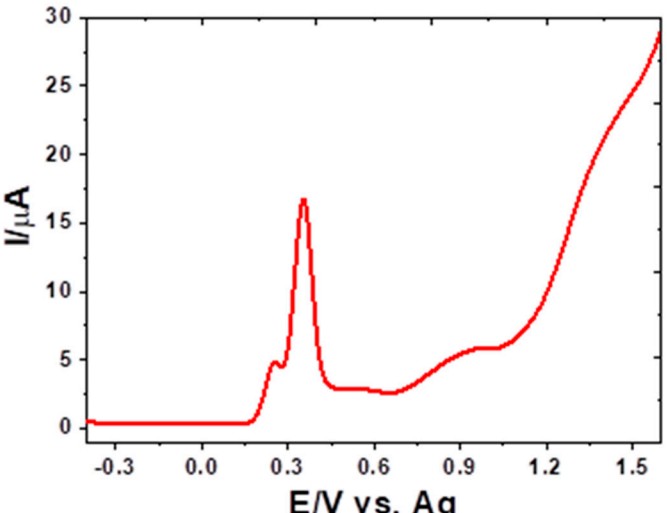

**Figure 6.** SWV of 68 µM keracyanin in 0.04 M B-R buffer (pH 2.2) at screen-printed carbon electrode (SPCE) at frequency 20 Hz.

The square wave voltammograms of KC using SWCNT-SPCEs (Figure 7A) and MWCNT-SPCEs (Figure 7B) present four anodic peaks representing the same hydroxyl groups as explained above. Figure 7A,B illustrates the first two peaks at ~0.1 mV and 0.45 mV, corresponding to the oxidation of the 3,4-hydroxyl groups of the B ring to form the corresponding *o*-quinone, and the remaining two peaks correspond to the oxidation of the hydroxyl groups of the A ring. It is worth noting that the modified screen-printed electrodes improve the sensitivity of the electrochemical response for KC oxidation. MWCNT-SPCEs show better sensitivity for KC oxidation than using SWCNT-SPCEs, illustrated by the high current values of the anodic peaks that correspond to the oxidation of the hydroxyl groups of the B ring and the oxidation of the hydroxyl groups of the A ring. Among the most interesting designs is the screen-printed carbon electrode (SPCE), which combines the working electrode (made from carbon-based material), the reference electrode, and the counter electrode in one single-printed substrate. A number of electrochemical measurements were conducted using an SPCE due to its advantages for microscale analysis. As a result of the inert nature of SPCE substrates, interferences during electrochemical measurements are avoided. Aside from its wide potential window, an SPCE is also inert, has a low background current [34], and is reasonably priced [35]. With disposable SPCE platforms, this study presents a promising electrochemical investigation and quantification protocol for KC, which offers several advantages, including reproducibility, ease of scaling, and low cost. Consequently, screen-printed carbon electrodes may be used to develop electrochemical sensors for analyzing KC.

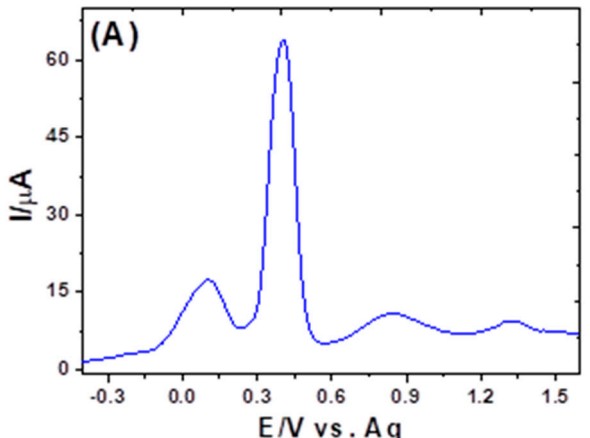 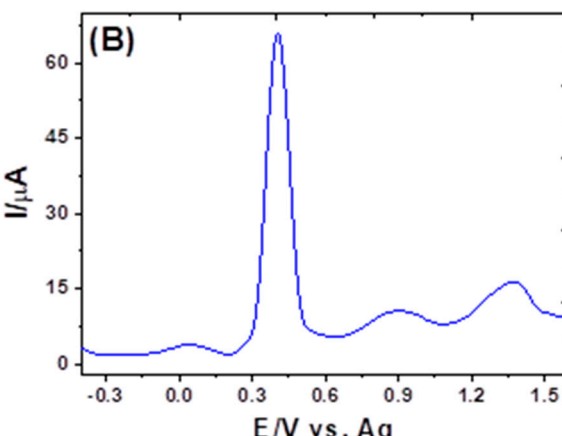

**Figure 7.** SWV of 68 µM keracyanin in 0.04 M B-R buffer (pH 2.2) at a frequency of 20 Hz on (**A**) SWCNT-SPCEs (**B**) MWCNT-SPCEs (supporting electrolyte background is subtracted from the two figures).

## 4. Conclusions

In the present study, keracyanin was characterized using glassy and screen-printed carbon electrodes using square wave and cyclic voltammetry techniques. The results of the experiments indicate that screen-printed carbon electrodes are sensitive to keracyanin detection. By understanding the electrochemical properties of keracyanin, we can better appreciate its antioxidant properties.

**Author Contributions:** Conceptualization, E.F.N.; methodology, E.F.N., A.G.S. and M.E.; software, E.F.N.; validation, E.F.N.; formal analysis, E.F.N.; investigation, E.F.N.; resources, E.F.N. and M.E. data curation, E.F.N., A.G.S. and M.E.; writing—original draft preparation, E.F.N.; writing—review and editing, E.F.N.; visualization, E.F.N.; supervision, E.F.N.; project administration, E.F.N.; funding acquisition, E.F.N. All authors have read and agreed to the published version of the manuscript.

**Funding:** This research received no external funding.

**Institutional Review Board Statement:** Not applicable.

**Informed Consent Statement:** Not applicable.

**Data Availability Statement:** The data presented in this study are available on request from the corresponding author.

**Conflicts of Interest:** The authors declare no conflict of interest.

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
