# Peer review of "Electrochemical Oxidation Profile of Anthocyanin Keracyanin on Glassy and Screen-Printed Carbon Electrodes"

_2673-3293, doi:10.3390/electrochem4020018_

Round 1

Reviewer 1 Report (Previous Reviewer 2)

I'm happy with the new changes that the authors made in the revised manuscript. 

The language and writing skills look okay to me. 

Author Response

The comments of Reviewer 1 have strengthened our communication, and we thank him/her for adding valuable comments. Thank you for accepting the work for publication.

Reviewer 2 Report (New Reviewer)

 Here are my suggestions/comments

1.       The introduction section needs to provide information about the electrochemical detection/study of anthocyanin keracyanin. It also requires more information about the significance of the electrochemical detection of anthocyanins and what electrodes have been used in the literature to study the electrochemical behavior of anthocyanins.

2.       The manuscript does not provide the background current in the absence of the target. It only provides the voltammetric current for detecting Keracyanin at the surface of GC and SPE electrodes.

3.       Figure 4 inset relates the peak area with the concentration of the Keracyanin, but it needs to be mentioned in the manuscript which peak and how the peak area is calculated.

4.       It is unclear whether the electrochemical oxidation is specific for Keracyanin or is generic for another related anthocyanin. I suggest performing the electrochemical detection with other anthocyanins and comparing the results.

5.       More detailed information about the effect of the pH and scan rate is necessary to explain the mechanism of electrochemical oxidation; however, the manuscript concludes the involvement of two electrons and two protons without any study. 

  1. The introduction section should provide a summary of the literature and clear objectives. Overall, it lacks coherent connections and reasoning/support.

Author Response

Response to Reviewer 2 Comments

Dear Prof. Dr. Editor of Electrochem,

My sincere thanks go out to you for your email regarding our communication. Furthermore, we are grateful for the valuable remarks and recommendations of the reviewer. Therefore, the communication has been revised to address all the comments raised by the reviewer. The following is our response to the comments "point-by-point":

The comments of Reviewer 2 have strengthened our communication, and we thank him/her for adding valuable comments.

  1. The introduction section needs to provide information about the electrochemical detection/study of anthocyanin Keracyanin. It also requires more information about the significance of the electrochemical detection of anthocyanins and what electrodes have been used in the literature to study the electrochemical behavior of anthocyanins. Our Response. Modifications were made to explain the electrochemical behavior of anthocyanin molecules in response to this comment.
  2. The manuscript does not provide the background current in the absence of the target. It only provides the voltammetric current for detecting Keracyanin at the surface of GC and SPE electrodes. Our Response. The background current in the absence of Keracyanin has been included in Figure 2.
  3. Figure 4 inset relates the peak area with the concentration of the Keracyanin, but it needs to be mentioned in the manuscript which peak and how the peak area is calculated. Our Response. As a result of your comment, the unit of peak area has been corrected, and our explanation of how peak area is calculated has been modified.
  4. It is unclear whether the electrochemical oxidation is specific for Keracyanin or is generic for another related anthocyanin. I suggest performing the electrochemical detection with other anthocyanins and comparing the results. Our Response. Thank you very much for your comment. The current study examines the electrochemical oxidation profile of Keracyanin, a member of the Anthocyanins family.
  5. More detailed information about the effect of the pH and scan rate is necessary to explain the electrochemical oxidation mechanism; however, the manuscript concludes the involvement of two electrons and two protons without any study. Our Response. As reported in previous publications (Electroanalysis 2012, 24, 1576–1583 and Electroanalysis 2018, 30, 1714-1722), it has been shown that the catechol groups in the B-ring of the molecule can be oxidized to ortho-quinone by losing two electrons and two protons. As a result, polyphenols oxidize electrochemically to produce the corresponding o-quinone using a two-electron two-proton reaction. A new experiment has been added to demonstrate the effect of pH on the electrochemical oxidation of Keracyanin. The text was modified in response to this comment.
  6. The introduction section should summarize the literature and clear objectives. Overall, it needs more coherent connections and reasoning/support. Our Response. The introduction section has been revised to provide better connections and reasoning/support.

There is no doubt that the reviewer contributed valuable comments to the paper, and we hope that our responses will be convincing to the reviewer and suitable for publication in Electrochem.

Thank you, and best wishes,

Emad F. Newair 

Reviewer 3 Report (New Reviewer)

The manuscript by Newair et al. deals with the electrochemical oxidation profile of Anthocyanin Keracyanin on glassy and screen-printed carbon electrodes. Authors have utilized CV and SWV to study the electrochemical response of KC oxidation. Further, authors found that the SPCE has better sensitivity than the GCE. The short manuscript is well written, however, authors should consider adding electrochemical equations and experimental schematic as it helps in adding more clarity to the manuscript. The manuscript can be published after addressing some minor points indicated below:

Pg1, ln22, authors should consider including relevant references on the previous works of electrochemical oxidation of anthocyanins and what is unique about authors current work

Pg 2, ln 44, authors may wish to clarify why a pH value of 2.2 was used for their study?

Fig 2, authors should comment on what is the minimum current that could be measured on their potentiostat (i.e., resolution) and what is the confidence. Further, authors should clarify their claims on Pg 2, ln80, as I am not able to see any shoulders in Fig. 2. And authors should consider adding a background CV scan without KC ions in the electrolyte

Pg 2, ln77, authors may wish to elaborate the pH Dependence on the oxidation potentials of Anthocyanins. Further, authors should comment on the significance for choosing 100 mV/s as the scan rate and what is the dependence of scan rate on electrochemical oxidation of KC. Additionally, authors must comment on the surface area of the glassy carbon electrode?

Pg 3, ln 111, authors should elaborate on the significance for choosing 68um as the KC concentration and how it affects the oxidation potential

Pg 3, ln 120, authors should clarify in the text that Fig 4 refers to adsorptive square wave voltammetry as it is unclear

Fig. 3, authors must comment on the difference in current measured between CV and SWV, i.e., almost 10x increase in current is observed using SWV when compared to CV. Further, authors may wish to clarify what is the electrochemical potential window for their electrolyte, i.e., at which point electrolyte starts to break down?

Fig, 5, authors should consider adding SWV of the background electrolyte without any KC electrolyte

Pg 5, ln 196, authors, should explain how the sensitivity of KC oxidation increases with SPCE over GCE for a stronger impact on the article

Author Response

Response to Reviewer 3 Comments

Dear Prof. Dr. Editor of Electrochem,

My sincere thanks go out to you for your email regarding our communication. Furthermore, we are grateful for the valuable remarks and recommendations of the reviewer. Therefore, the communication has been revised to address all the comments raised by the reviewer. The following is our response to the comments "point-by-point":

The comments of Reviewer 3 have strengthened our communication, and we thank him/her for adding valuable comments.

  1. The short manuscript is well written; however, authors should consider adding electrochemical equations and an experimental schematic as it helps in adding more clarity to the manuscript. Our Response. We greatly appreciate your suggestion. As a result of this comment, the electrochemical oxidation mechanism of anthocyanin molecules has been added, the text was modified to include the oxidation scheme.
  2. Pg1, ln22, authors should consider including relevant references on the previous works of electrochemical oxidation of anthocyanins and what is unique about the authors’ current work. Our Response. Modifications were made to explain in greater detail how the work is important and novel. We modified the text by adding references, results, and discussion to clarify these important points.
  3. Pg 2, ln 44, authors may wish to clarify why a pH value of 2.2 was used for their study. Our Response. A new experiment has been added to demonstrate the effect of pH on the electrochemical oxidation of Keracyanin. The text was modified in response to this comment.
  4. Fig 2, authors should comment on the minimum current that could be measured on their Potentiostat (i.e., resolution) and the confidence. Further, the authors should clarify their claims on Pg 2, ln80, as I cannot see any shoulders in Fig. 2. Authors should consider adding a background CV scan without KC ions in the electrolyte. Our Response. Thank you for this valuable suggestion. The text has been updated to include the following information: The current resolution is 0.0003 %, and the current accuracy is 0.2 % of the current range. Furthermore, Figure 2 has been modified to include the background current in the absence of Keracyanin.
  5. Pg 2, ln77, authors may wish to elaborate on the pH Dependence on the oxidation potentials of Anthocyanins. Further, the authors should comment on the significance of choosing 100 mV/s as the scan rate and the dependence of the scan rate on the electrochemical oxidation of KC. Additionally, authors must comment on the surface area of the glassy carbon electrode. Our Response. At a scanning rate of 100 mV s-1, the general voltammogram of Keracyanin was observed to determine its CV behavior, but cyclic voltammetry is less sensitive than square wave voltammetry. SWV is advantageous over cyclic voltammetry since it consumes fewer electroactive species, analyzes more rapidly, and does not pose as many problems associated with electrode poisoning. Thus, we intend to conduct a complete study using the SWV of Keracyanin. The text has been modified to include the surface area of the glassy carbon electrode. In addition, a new figure was added (Figure 4B) illustrating the effect of pH on the oxidation potential.
  6. Pg 3, ln 111, authors should elaborate on the significance of choosing 68µm as the KC concentration and how it affects the oxidation potential. Our Response. A concentration of 68m KC was used in Figure 2 to visualize the general voltammogram of KC; however, Figure 5 examined the effect of KC concentration to estimate the detection limit.
  7. Pg 3, ln 120, the authors should clarify in the text that Fig 4 refers to adsorptive square wave voltammetry as it is unclear. Our Response. Thank you so much for this essential suggestion. The text was modified in response to this comment.
  8. 3, authors must comment on the difference in current measured between CV and SWV, i.e., almost a 10x increase in current is observed using SWV when compared to CV. Further, authors may wish to clarify the electrochemical potential window for their electrolyte, i.e., at which point does the electrolyte break down? Our Response. In response to this comment, the text was modified to illustrate the sensitivity of SWV. Regarding the potential window of the B-R buffer, we detect the beginning of O2 or H2 evolution for the electrochemical system without the target molecule (KC), and the B-R buffer supporting electrolyte provides a very stable potential window of more than 2V, during which the target molecule is likely to oxidize.
  9. Fig, 5, authors should consider adding SWV of the background electrolyte without any KC electrolyte. Our Response. Yes, we have done that, and the background has been subtracted from the SWV. There has been a modification to the figure’s caption to make it clearer.
  10. Pg 5, ln 196, the authors should explain how the sensitivity of KC oxidation increases with SPCE over GCE for a stronger impact on the article. Our Response. Thank you so much for this important suggestion. The text was modified in response to this comment.

There is no doubt that the reviewer contributed valuable comments to the paper, and we hope that our responses will be convincing to the reviewer and suitable for publication in Electrochem.

Thank you, and best wishes,

Emad F. Newair 

Round 2

Reviewer 2 Report (New Reviewer)

Since the oxidation mechanism is not from the current work, please consider removing this statement from the conclusion section. "Two electrons and  two protons are involved in a redox reaction driven by the hydroxyl groups attached to the catechol moiety"

Although writing is good grammatically, authors can make it better by using better flow. for e.g. in line 201, the authors have mentioned the linear calibration curve, without defining what they are plotting. 

Author Response

In response to the comments of Reviewer 2:

  1. Since the oxidation mechanism is not from the current work, please consider removing this statement from the conclusion section. "Two electrons and two protons are involved in a redox reaction driven by the hydroxyl groups attached to the catechol moiety". Our response. The statement was removed from the text.
  2. Although the writing is good grammatically, authors can make it better by using better flow, e.g., in line 201, the authors mentioned the linear calibration curve without defining what they are plotting. Our response. The text was modified in response to this comment to be more obvious. 

This manuscript is a resubmission of an earlier submission. The following is a list of the peer review reports and author responses from that submission.

Round 1

Reviewer 1 Report

Dear Authors,

In my opinion you need to make some changes to improve your manuscript:

1. Introduction

- line 29: you should specify that the O-H bonds related with the antioxidant effect are O-H attached with the aromatic ring (not the glycoside OH).

- the phrase "Therefore, it is critical to understand the electrochemical properties of anthocyanin Keracyanin chloride" is too dramatic

-"It is reported that the B-ring of flavonoids is more oxidizable than the A-ring of resorcinol" - A and B rings are flavonoid rings. You should say only A or B ring as it is presented in figure 1. 

2. Material and methods

- you should present pH of the BR buffer obtained after adding NaOH

- water/ethanol mixture - how were prepared?

- "The electrode was immediately electro- 71 chemically measured after casting 200 μL of a Keracyanin chloride solution onto it." - Why?

- for the explanation in lines 78 -81 (page 2) you present references that are your articles. For this observations you should add additional references.

The paper has no conclusions.

The manuscript ends with the phrase: "Therefore, applicable, inexpensive, and simple electrochemical sensors can be developed for KC analysis in food."  You tested only one solution 68 mM in 0.04 M BR buffer (pH 2.2) .

Reviewer 2 Report

The manuscript presented by Newair and Shehata focuses mostly on the electrochemical properties of keracyanin chloride (KC) in an aqueous buffer solution. I found this work has very narrow objectives and does not look suitable for a broad audience in the chemistry community. Therefore, I do not recommend publication of this manuscript in its current form.

1. The authors should showcase a big picture of this project and expand it more. The results look premature.

2. The discussion suddenly jumped into KC without prior background and proper rationalization. 

3. The authors should discuss the broad impact of this project. 

Reviewer 3 Report

In this work the authors described the electrochemical behavior of Anthocyanin Keracyanin at different carbon-based types electrodes using cyclic voltammetry and Square wave voltammetry. The manuscript is clear, but the following issues should be addressed:

1. the sentence: Therefore, it is 31 critical to understand the electrochemical properties of anthocyanin Keracyanin chloride 32 (Cyanidin-3-O-rutinoside chloride, Figure 1) to better understand its antioxidant proper-33 ties. It is reported that the B-ring of flavonoids is more oxidizable than the A-ring of res-34 orcinol [19], should be reformulated.

2. what is the volume ratio for water/ethanol mixture?

3. in section 2.3 please correct the dimension of alumina powder, i.e. microns instead of m

4. in section 2.4 , row 69, avoid "bare" when refer to screen printed electrude; please correct also in section 3.2 at the rows 122 and 125.

5. in section 3.1 row 75 the correct concentration is 68 μM Keracyanin; the same for row 109.

6. to support the potential application of this method in quantification of Keracyanin I consider is mandatory a linear response within a concentration range of the analyte.

7. why the authors choose the pH 2.2 for their study?
